Evolutionary history determines how plant productivity responds to phylogenetic diversity and species richness

Genung Mark A. mgenung@utk.edu
Schweitzer Jennifer A.
Bailey Joseph K.
Department of Ecology and Evolutionary Biology, University of Tennessee–Knoxville , Knoxville, TN , USA
Bezemer T. Martijn
Electronic publication date: 2014 Mar 13
Publication date: 2014
Volume: 2
Electronic Location ID: e288
Received 2013 Oct 4; Accepted 2014 Feb 4
Copyright: © 2014 Genung et al.
Copyright year: 2014
Copyright holder: Genung et al.
License: This is an open access article distributed under the terms of the Creative Commons Attribution License, which permits unrestricted use, distribution, and reproduction in any medium, provided the original author and source are credited.
License URL: https://creativecommons.org/licenses/by/3.0/

Keywords: Ecosystem function, Phylogeny, Biodiversity, Evolutionary history, Species richness, Species interactions

Funding: National Science Foundation DEB-0743437 The Australian Research Council FT0991727 The Department of Ecology and Evolutionary Biology at the University of Tennessee, the National Science Foundation (DEB-0743437) and The Australian Research Council (FT0991727) provided financial support to the authors. The funders had no role in study design, data collection and analysis, decision to publish, or preparation of the manuscript.

==============================
The relationship between biodiversity and ecosystem function has received a great deal of attention in ecological research and recent results, from re-analyses, suggest that ecosystem function improves with increases in phylogenetic diversity. However, many of these results have been generalized across a range of different species and clades, and plants with different evolutionary histories could display different relationships between biodiversity and ecosystem function. To experimentally test this hypothesis, we manipulated species richness and phylogenetic diversity using 26 species from two subgenera of the genus Eucalyptus (subgenus Eucalyptus and subgenus Symphyomyrtus). We found that plant biomass (a measurement of ecosystem function) sometimes, but not always, responded to increases in species richness and phylogenetic diversity. Specifically, Symphyomyrtus plants showed a positive response while no comparable effect was observed for Eucalyptus plants, showing that responses to biodiversity can vary across different phylogenetic groups. Our results show that the impacts of evolutionary history may complicate the relationship between the diversity of plant communities and plant biomass.

Introduction

Biodiversity is fundamental to ecosystem services (Millennium Ecosystem Assessment, 2005), and it has been shown that ecosystem function generally improves with increasing species richness (Naeem et al., 1996; Hooper & Vitousek, 1997; Hector et al., 1999; Troumbis et al., 2000; Tilman et al., 2001). Similarly, losses of species from communities can negatively affect ecosystem services (e.g., Cardinale et al., 2006; Cardinale et al., 2012; Tilman, Reich & Isbell, 2012). The most common mechanistic explanations for the relationship between biodiversity and ecosystem function (reviewed in Hooper et al., 2005), namely complementarity (e.g., Trenbath, 1974; Vandermeer, 1992) and facilitation (e.g., Callaway, 1995; Valiente-Banuet & Verdu, 2007), are driven by particular plant traits. The importance of traits suggests that evolutionary history, which generates the mechanistic basis for biodiversity effects (trait diversity), may play a large role in influencing ecosystem function (Srivastava et al., 2012), and in fact recent re-analyses of studies of species-richness experiments have shown that phylogenetic diversity (i.e., the level of phylogenetic relatedness among individuals within a community) can be a better predictor of ecosystem function than species richness (Cadotte, Cardinale & Oakley, 2008; Cadotte et al., 2009; see Cadotte, 2013 for an experimental approach to the same issues). While convergent evolution may give rise to similar traits in lineages that have different evolutionary histories, in general, individuals with different evolutionary histories may respond in different ways to increases in phylogenetic diversity. In other words, because different evolutionary histories give rise to different traits, it is possible that the effects of species richness on ecosystem function will vary across different phylogenetic groups (i.e., clades or other distributions).

Our understanding of how biodiversity drives ecosystem function has been advanced by the recognition that greater phylogenetic diversity may lead to greater ecosystem function, through mechanisms related to complementarity and facilitation. For example, Flynn et al. (2011) showed that both functional diversity and phylogenetic diversity predicted plant productivity despite the fact that the correlation between these two metrics was weak or non-existent. The conclusion that functional and phylogenetic diversity separately affect ecosystem function suggests that it may be difficult to identify and measure all traits that are associated with plant productivity or other ecosystem processes, and potentially indicates the existence of complex relationships between plant traits, species richness, and phylogenetic relatedness. Cadotte (2013) helped to illuminate the interactions between species richness and phylogenetic relatedness by showing that the effects of species richness on the productivity of plant mixtures depended on the amount of phylogenetic diversity present in the mixtures. Despite this work, little is known about how phylogenetic diversity interacts with evolutionary history (i.e., evolutionary forces that drive cladogenesis) to impact the traits that affect plant productivity and community interactions. Studies that manipulate species richness, phylogenetic diversity and phylogenetic relatedness within and among taxa can determine the extent to which each of these factors influence ecosystem function.

Eucalypts are ideal for experimentally addressing the relationship between phylogenetic diversity and ecosystem function because of the high degree of variation in relatedness and the co-occurrence of many of these species in the field. We used 26 native Eucalyptus species in two subgenera, Eucalyptus (10 species) and Symphyomyrtus (16 species), which are distributed in various habitats across a range of elevations in Tasmania, Australia, to examine how phylogeny and biodiversity interact to influence plant growth traits and survival. Hereafter, the terms Eucalyptus and Symphyomyrtus will refer to subgenus identities within the genus Eucalyptus. Individual plants were planted in one of three “species richness” treatments—monocultures, three-species mixtures, or six-species mixtures. We created two mixture types—one from which we randomly drew species from only one subgenus (hereafter “within-subgenus mixtures”, for both Eucalyptus and Symphyomyrtus), and another treatment for which we drew constituents from both subgenera (hereafter “between-subgenera mixtures”). We use two metrics to represent “biodiversity”: (1) “species richness” (SR), which is the number of species present in each pot and (2) “phylogenetic diversity” (PD), as a categorical variable with three levels (monocultures, within-subgenus mixtures, and between-subgenera mixtures). This approach allows us to qualitatively compare the main effects of SR and PD, as well as to test for interactions between these factors and evolutionary history (i.e., subgenus identity). We hypothesized that plants from subgenus Symphyomyrtus would show a stronger response to increases in biodiversity (either SR or PD) because this subgenus has faster growth rates (in the absence of herbivores; Stone, Simpson & Gittins, 1998) and more foliar N (Wallis, Nicolle & Foley, 2010) (suggesting adaptation to higher nutrient environments; Hobbie, 1992), meaning that plants in this subgenus may benefit more from potential niche partitioning for nutrient acquisition in mixtures.

Materials and methods

We established an experiment that used 26 (of 29 total) Tasmanian eucalypt species, planted in species monocultures and mixtures. Of the 26 species, 16 are in subgenus Symphyomyrtus and 10 are in subgenus Eucalyptus. There are 3 series (a phylogenetic designation smaller than subgenus) within Symphyomyrtus and 2 within Eucalyptus (see Senior et al., 2013 for species lists within each series). These phylogenetic classifications are based on a framework created by Brooker (2000) with recent molecular data (Diversity Arrays Technology or DArT, Jaccoud et al., 2001) supporting the subgenus- and series-level classifications (McKinnon et al., 2008; Steane et al., 2011; Senior et al., 2013). Seeds from 1–6 individuals, from 1–3 populations of each species were acquired from Forestry Tasmania. We vernalized seeds for 30 days in water (with a drop of dish detergent) and stored the seeds at 4°C. Seed from each species were then germinated on the soil surface in separate flats and kept under uniform, moist conditions in a greenhouse. After 10 weeks, we transplanted the seedlings into monoculture or mixture treatments; more information on the treatments is given later in the methods. At this point seedlings had only barely emerged from the soil and had, on average, one leaf. The height of these seedlings was less than 5 cm. The seedlings were transplanted into a standard commercial potting mix (Nutricote Grey; Langley Australia Pty Ltd., Welshpool, Western Australia), with 19:2.6:10 (N:P:K) granular fertilizer applied at a rate of 3 kg/m3. Each large 30 L (diameter ∼35 cm) pot contained an equal density of six seedlings. Within each pot, the six individual plants were planted randomly in a circular pattern, after species assignments. The pots of seedlings were grown for 50 days in randomized positions in the glasshouse and then the pots were moved to a fenced, outdoor location on the University of Tasmania campus. Plants were watered daily, and evenly, with automatic sprinkler systems. Weeds in the pots were occasionally removed to minimize their effects on growth of the seedlings.

We measured plant height (cm), stem diameter (mm), and plant survival on individual plants over the course of two months (19 July through 13 September 2011). During this time, temperatures typically range from 5° to 15°C and rainfall is around 50 mm per month. We quantified the death of individual seedlings, but mortality was ∼3% and therefore all pots were utilized in the statistical analyses. We measured stem diameter at the base of the plants, just above the soil surface. Our analyses use the height and stem diameter measurements from the final date (13 September 2011). We stress that our results should be interpreted in accordance with the timescale of the study (described above); for example, the growth rates of subgenus Eucalyptus and subgenus Symphyomyrtus are known to vary with ontogeny (and also between greenhouse and field conditions) (Duff, Reid & Jackson, 1983). We used height and stem diameter measurements from other Eucalyptus plants to construct an allometric equation that predicted total (combined above- and belowground together) biomass. We grew three individuals of each of the 26 species and sampled one individual from each species at three different sampling dates to obtain a range of height and stem diameter measurements (height ranged from 7.2 to 105.0 cm; stem diameter ranged from 1.84 to 10.04 mm). This allometric equation explained 86.2% of the variation in total biomass, and is given here: Total biomass (g) = (Height (mm) ∗ 0.0129) + (Stem Diameter (mm) ∗ 2.8207) + (((Height (mm) − 577.821) ∗ (Stem Diameter (mm) − 5.554)) ∗ 0.0042) − 13.796. Plants that did not survive were given “blank” values for height, stem diameter, and biomass, which were not analyzed.

Treatments included species monocultures for all 26 species (n = 2 for 52 monoculture pots or 312 plants) and different types of mixture pots. Mixtures (n = 34 pots or 204 plants; combined with monocultures this yields a total of 86 pots or 516 plants of which 502 survived the initial transplanting and were used in the analyses) were created through a random draw of species to create pots with either three or six species; some species mixtures contained plants from only one subgenus, while other species mixtures contained plants from both subgenera (Eucalyptus and Symphyomyrtus). Three-species pots that included species from both subgenera (by necessity) included 4 individuals of a given subgenus and 2 from the other subgenus. Due to the random sampling of species to compose mixtures, six-species pots that included species from both subgenera sometimes included 3 species from a given subgenus and 3 species from the other subgenus (i.e., a 3/3 split), and sometimes a 4/2 split. There were no pots that included a 5/1 split. This process created two “diversity” treatments within the same common garden experiment—a species richness (SR) treatment and a phylogenetic diversity (PD) treatment. By definition, adding species to a species monoculture will increase both SR and PD, and the results should be interpreted in this context (i.e., increases in SR cannot necessarily be interpreted as occurring independently of increases in PD). The levels of PD are: species monocultures, within-subgenus mixtures, and between-subgenera mixtures. Although monocultures and within-subgenus mixtures both contain representatives of only one subgenus, within-subgenus mixtures still display more phylogenetic variation than monocultures because of variation in species composition. Within-subgenus mixtures could have either 3 or 6 species, but those species were required to come from one subgenus. Between-subgenera mixtures could have either 3 or 6 species, but both subgenera (Eucalyptus and Symphyomyrtus) were required to be represented.

Of the 502 plants, 312 (303 survived) were in monoculture and 190 (185 survived) were in some type of mixture. For SR, 129 (126 survived) plants were in 3-species mixtures, and 61 (59 survived) were in 6-species mixtures. For PD, 167 plants (164 survived) were in within-subgenus mixtures, and 23 (21 survived) were in between-subgenus mixtures. If SR and PD are considered factorially, then 117 plants (115 survived) were in “3-species, within-subgenus” mixtures, 12 (11 survived) were in “6-species, between-subgenus” mixtures, 50 (49 survived) were in “3-species, within-subgenus” mixtures, and 11 (10 survived) were in “6-species, between-subgenus” mixtures. Our manipulations of PD are broad and categorical; we use “subgenus richness” as a simple measure of the phylogenetic diversity present in the community and our results should be interpreted in this light. Additionally, Australian eucalypt species frequently co-occur, including con-subgeneric species pairs (Parsons & Rowan, 1968; Rogers & Westman, 1979) and distribution maps indicate that most species are capable of occupying the same environments (Williams & Potts, 1996) suggesting species interactions among Eucalyptus are present in natural systems. While 6-species mixtures are likely not the norm in natural settings, we employed 6-species mixtures to test the range of variation that is possible (in terms of response to diversity in neighboring plants) for plants within genus Eucalyptus.

Statistical methods

We used two REML (restricted maximum likelihood) models, one for SR and one for PD, to determine whether subgenus identity and biodiversity affected plant performance. The fixed effects for this model were subgenus identity (Eucalyptus or Symphyomyrtus), level of diversity (monocultures, 3-species mixtures, and 6-species mixtures for the SR model; monocultures, within-subgenus mixtures, and between-subgenera mixtures for the PD model), and the interaction of subgenus identity and level of diversity. The random effects were pot identity (to control for potential variation across pots) and species identity (nested within subgenus identity; to account for differences across the 26 species). The response variables were plant height, stem diameter, survival, and biomass, at the individual level. We recognize that a 3-way model including SR, PD, and subgenus identity would be useful, but we do not have the sample size (specifically, in the 6-species SR ∗ between-subgenera PD combination) to run a robust 3-way model. Therefore, we stress that our two models should be used to evaluate the effect of SR and subgenus identity or PD and subgenus identity, but should not be used to quantitatively compare how SR and PD affect plant performance. For PD, which was a categorical variable, we used post-hoc contrasts to examine pairwise differences between levels of PD (monocultures, within-subgenus mixtures, and between-subgenus mixtures). Analyses were carried out in JMP 9.0 and when multiple comparisons were made we controlled the False Discovery Rate (Verhoeven, Simonsen & McIntyre, 2005; Pike, 2011) at 0.05. The REML output from JMP 9.0 includes the degrees of freedom that provide the closest match between the F distribution and the distribution of the test statistic (Kenward & Roger, 1997).

Because the species composition of the non-monoculture pots had been randomly sampled, we needed to account for the composition of species at each level of species richness (i.e., three- and six-species mixtures) and phylogenetic diversity (i.e., within-subgenus and between-subgenera mixtures). This is because random sampling resulted in variable species composition across different levels of diversity (either SR or PD) and not accounting for this would confound species identity and composition with diversity. For each response variable, we assembled a list of the mean response variable for each species when grown in monoculture, and also assembled a list of the proportional representation of each species in each mixture type (3-species mixtures, 6-species mixtures, within-subgenus mixtures, between-subgenera mixtures). For each species, we then multiplied the mean response variable by proportional representation and summed the values for all species. Finally, we compared the expected values for our random draw of species (calculation described above) with the expected values for diversity treatments in which all species were equally represented (following the methods described above, this expected value is equal to the monoculture mean because the proportional representation of all species would be equivalent under these conditions). We found that the random sampling of species had minimal effects (average of 0.8% and maximum of 2.4% difference) on expected values in the mixture pots; a table comparing the difference between “random draw” expectations and “equal representation” expectation for all “trait by diversity level” combinations is attached as a supplementary document (Appendix S1). With respect to SR and PD, the magnitudes of the effects we observe are much greater those described above. Because of this, we argue that our “subgenus ∗ diversity level” models (described in the preceding paragraph) are an acceptable way to analyze this data.

Results

As predicted, evolutionary history (subgenus-level differences) mediated the relationship between biodiversity and plant performance. In analyses that classified mixture pots according to species richness (SR), we found that SR and subgenus identity interacted to affect stem diameter, survival, and biomass (Table 1, Fig. 1). For stem diameter and biomass, the interactions were driven by a positive response of plants within Symphyomyrtus to increasing SR (stem diameter + 28% and biomass + 28%, in 6-species mixtures relative to monocultures), compared with a less-pronounced positive response of plants within Eucalyptus (stem diameter + 6% and biomass + 6%). For survival, the interaction was driven by a negative response of plants within Eucalyptus (survival −12% in 6-species mixtures relative to monocultures) compared to a slightly positive response for plants within Symphyomyrtus (survival + 3%).

Figure 1 The effects of species richness on plant biomass and survival are dependent on subgenus identity.

The species richness manipulation included monocultures (species richness = 1), 3-species mixtures, and 6-species mixtures. Plant height (A), stem diameter (B), and biomass (D) responded positively to increasing species richness, but only in subgenus Symphyomyrtus (open circles and dashed lines). Survival (C) responded negatively to increasing species richness, but only in subgenus Eucalyptus (closed circles and solid lines).

Table 1 Species richness, phylogenetic diversity, and subgenus identity affect plant productivity and survival.

“Species richness” is a continuous variable with three levels (1, 3, and 6). “Phylogenetic diversity” is a categorical variable with three levels (species monocultures, within-subgenus mixtures, between-subgenera mixtures). Within-subgenus and between-subgenera mixtures can include 3 or 6 species; the difference is whether those species come from one subgenus or two. The term subgenus describes differences between plants within subgenus Eucalyptus and subgenus Symphyomyrtus. Plant height and stem diameter were continuous responses, and we used REML models with species identity and pot number as random effects. “df Den.” is an abbreviation for denominator degrees of freedom, and shows the degrees of freedom that causes the distribution of the test statistic to most closely match the F distribution. Bold, italicized p-values are significant at α = 0.05.

Species richness model
(N = 86 pots)	df Den.	F	p	df Den.	F	p	
	Plant height (485 plants)	Stem Diameter (485 plants)	
Species Richness (SR)	56.12	9.029	0.004	63.63	12.300	0.001	
Subgenus	23.72	0.834	0.370	24.82	2.407	0.134	
SR ∗ Subgenus	66.02	2.502	0.119	78.09	5.657	0.020	
	Survival (505 plants)	Biomass (485 plants)	
Species Richness (SR)	69.77	2.572	0.113	63.65	12.193	0.001	
Subgenus	21.92	2.528	0.126	24.72	2.269	0.145	
SR ∗ Subgenus	96.49	6.276	0.014	77.64	5.668	0.020	
Phylogenetic diversity model
(N = 86 pots)	df Den.	F	p	df Den.	F	p	
	Plant Height (485 plants)	Stem Diameter (485 plants)	
Phylogenetic diversity (PD)	57.05	8.588	0.005	63.07	10.30	0.002	
Subgenus	23.63	0.804	0.379	24.52	2.360	0.137	
PD ∗ Subgenus	99.66	4.486	0.037	123.15	7.850	0.006	
	Survival (505 plants)	Biomass (485 plants)	
Phylogenetic diversity (PD)	68.35	8.13	0.006	63.05	10.27	0.002	
Subgenus	21.40	1.99	0.173	24.43	2.22	0.149	
PD ∗ Subgenus	159.84	6.18	0.014	120.94	8.11	0.005	

In analyses that classified mixture pots according to phylogenetic diversity (PD), we detected interactions between PD and subgenus identity for height, stem diameter, survival, and biomass (Table 1, Fig. 2). Pairwise, post-hoc contrasts indicated that the PD ∗ subgenus interaction terms for height, stem diameter and biomass were driven by differences in Symphyomyrtus performance between monocultures and within-subgenus mixtures (Table 2). The effects on height, stem diameter and biomass were positive for Symphyomyrtus (height + 18% in within-subgenus mixtures relative to monocultures, and + 18% in between-subgenera mixtures relative to monocultures; stem diameter +25%/+28%; biomass +25%/+28%) and nearly neutral for Eucalpytus (height −4%/+6%; stem diameter −2%/+6%; biomass −2%/+6%). Pairwise, post-hoc contrasts indicated that, for survival, the PD ∗ subgenus interaction was driven by differences in Eucalyptus survival in between-subgenus mixtures relative to monocultures and within-subgenus mixtures (Table 2). Survival of Eucalyptus plants was lower in between-subgenus mixtures relative to monocultures (−4%) and within-subgenus mixtures (−12%); survival of Symphyomyrtus plants responded less strongly (−2% in between-subgenus mixtures, and + 3% in within-subgenus mixtures, relative to monocultures).

Figure 2 The effects of phylogenetic diversity on plant biomass and survival are dependent on subgenus identity.

The phylogenetic diversity manipulation included monocultures, within-subgenus mixtures (with species richness of either 3 or 6), and between-subgenera mixtures (with species richness of either 3 or 6). Plant height (A), stem diameter (B), and biomass (D) responded positively to increasing phylogenetic diversity, but only in subgenus Symphyomyrtus. Survival (C) responded negatively to increasing phylogenetic diversity, but only in subgenus Eucalyptus. Letters indicate the results of pairwise contrasts within each subgenus (uppercase for Eucalyptus, lowercase for Symphyomyrtus); groups with different letters are significantly different; to account for multiple tests, we controlled the False Discovery Rate at 0.05.

Table 2 Pairwise contrasts show that different levels of phylogenetic diversity affect plant traits.

The phylogenetic diversity manipulation included monocultures, within-subgenus mixtures (with species richness of either 3 or 6), and between-subgenera mixtures (with species richness of either 3 or 6). The first column describes the two levels of phylogenetic diversity being compared; the next four columns show p-values for different plant traits. The first set of contrasts tests differences for the main effect of phylogenetic diversity. The second and third sets of contrasts test differences for the interactive effect of subgenus identity by phylogenetic diversity. Bold, italicized p-values are significant at α = 0.05 and (+) symbols show which category showed a higher mean value for the listed plant traits.

Contrast description	Plant height	Stem diameter	Survival	Total biomass	
Monoculture & Within-Subgenus (+)	0.004	<0.001	0.701	<0.001	
Monoculture (+) & Between-Subgenus	0.217	0.605	<0.001	0.571	
Within-Subgenus (+) & Between-Subgenus	0.968	0.268	<0.001	0.301	
Eucalyptus Monoculture & Within-Subgenus	0.831	0.708	0.445	0.760	
Eucalyptus Monoculture (+) & Between-Subgenus	0.516	0.879	<0.001	0.911	
Eucalyptus Within-Subgenus (+) & Between-Subgenus	0.580	0.762	<0.001	0.815	
Symphyomyrtus Monoculture & Within-Subgenus (+)	<0.001	<0.001	0.781	<0.001	
Symphyomyrtus Monoculture & Between-Subgenus	0.245	0.325	0.277	0.312	
Symphyomyrtus Within-Subgenus & Between-Subgenus	0.574	0.165	0.242	0.178	
Eucalyptus Monoculture (+) & Symphyomyrtus Monoculture	0.141	0.019	0.907	0.020	
Eucalyptus Within-Subgenus & Symphyomyrtus Within-Subgenus	0.701	0.608	0.299	0.585	
Eucalyptus Between-Subgenus & Symphyomyrtus Between-Subgenus (+)	0.599	0.666	0.003	0.665	

Discussion

Both measures of biodiversity (“species richness” and “phylogenetic diversity”) had impacts on plant communities. The phylogenetic identity of different groups determined whether increases in biodiversity had a significant effect on a given response variable. For plant height, stem diameter, and biomass, we generally observed a positive response to SR for species within the Symphyomyrtus lineage compared with no response for species within the Eucalyptus lineage (Figs. 1 and 2). For plants within Symphyomyrtus (but not Eucalyptus), plant height, stem diameter and biomass were greater in within-subgenus mixtures than in monocultures, but between-subgenera mixtures did not differ from either monocultures or within-subgenus mixtures. These results suggest an inconsistent relationship between PD and plant performance, as increasing dissimilarity from monocultures to within-subgenus mixtures increased plant performance but this positive effect disappeared in between-subgenera mixtures. For plant survival, we found that plants within Eucalyptus grown in mixtures (either SR or PD) had higher mortality than plants within Eucalyptus grown in species monocultures. Taken together, our results are a proof-of-concept that even phenotypically similar, closely related phylogenetic groups may show different responses to varying levels of biodiversity.

Subgenus identity determined how all response variables (plant height, stem diameter, survival, and biomass) responded to growing in mixtures rather than monocultures (see interaction terms in Table 1), indicating that evolution has produced different relationships between biodiversity and plant biomass/survival. Moving past this result, the next interesting question involves understanding why these differences exist and whether they are linked with trait divergence. A number of factors (e.g., degree of sympatry, competition, etc.) can affect rates of trait divergence and there have been calls for alternative evolutionary models that can better apply macro-evolutionary patterns to ecological questions (Cadotte et al., 2009; Mouquet et al., 2012; Srivastava et al., 2012). One possible explanation for differences between Eucalyptus and Symphyomyrtus involves differences in landscape-level aggregation patterns; Eucalyptus species show more clustered distributions while the distributions of Symphyomyrtus species are more disjunct (Williams & Potts, 1996). Perhaps plants within Eucalyptus are less plastic than plants within Symphyomyrtus, limiting their distribution to certain regions and also limiting their ability to respond to different levels of diversity in the surrounding plant communities, but this is purely speculative. Additionally, species within Symphyomyrtus invest less in defense (Stone, Simpson & Gittins, 1998) and have more available foliar N (Wallis, Nicolle & Foley, 2010), two traits which suggest a fast-growth strategy that is adapted to higher-nutrient environments (Hobbie, 1992); perhaps this strategy enables plants within Symphyomyrtus to benefit more from potential niche partitioning, leading to more nutrient availability, in diverse pots. While we do not have the data to provide a mechanistic explanation of how evolution creates different relationships between biodiversity and plant biomass, it is clear that changes in SR and PD have different effects depending on the response variable in question and the phylogenetic identity of the plant on which the response was measured.

Because of the large range in genetic variation between within-subgenus and between-subgenera mixtures, we expected to find different results for plant biomass and survival when comparing these two types of pots. However, we did not observe greater biomass in two-subgenus pots compared with one-subgenus pots (see post-hoc tests in Fig. 2), suggesting that (1) within-subgenus interactions can create mixture effects of the same magnitude as between-subgenera interactions, (2) the relationship between biodiversity and ecosystem function may be more accurately represented by the interaction of evolutionary history and PD than by measures of diversity that ignore phylogenetic groups, and (3) some increases in PD do not result in increased ecosystem function. Darwin (1859) stated that closely related plants will compete more intensely; however, recent analyses have shown conflicting results which either support (e.g., Burns & Strauss, 2011) or contradict (e.g., Cahill, Lamb & Keddy, 2008; Kunstler et al., 2012) this hypothesis. For subgenus Eucalyptus, our productivity results do not indicate a relationship between phylogenetic relatedness within mixtures (i.e., within-subgenus mixtures vs. between-subgenera mixtures) and the intensity of competition between them, and our survival results suggest that, if anything, competition is greater between distantly related plants. For subgenus Symphyomyrtus, plants were more productive when growing with less-closely related plants, but this did not lead to an increase in survival. One possible explanation for these patterns is that two distantly related groups could demonstrate patterns of parallel evolution along similar environmental gradients, meaning that distantly related species that occupy similar niches could demonstrate large differences in phylogenetic diversity and yet be strongly competitive. This possibility also serves as an example for why more sophisticated, mathematical models linking phylogenies and ecology are needed (e.g., Cadotte et al., 2009; Mouquet et al., 2012; Srivastava et al., 2012 as mentioned above).

Consistent with previous studies (Naeem et al., 1996; Hooper & Vitousek, 1997; Hector et al., 1999; Troumbis et al., 2000; Tilman et al., 2001), our results show that SR can be positively correlated with increasing plant growth. While other studies (e.g., Tilman et al., 2001; Kunstler et al., 2012) have shown that different species respond differently to increases in the biodiversity of interacting species, our results build upon this established framework by focusing on groups of related species (i.e., subgenera) and experimentally showing that the interaction of evolutionary history and biodiversity (either SR or PD) can influence contemporary ecological processes. This interaction is potentially due to evolution driving different patterns of trait development in different phylogenetic groups, which affects the relative roles of competition and facilitation in mixtures comprised of different subgenera. Given that plant traits play a critical role in driving community and ecosystem-level effects of species richness (Cadotte, Cardinale & Oakley, 2008; Flynn et al., 2011), and that trait diversification rates can vary tremendously across different phylogenetic groups (Ackerley, 2009), these results may provide a first step for understanding how different evolutionary histories may interact with patterns of biodiversity to shape species interactions. In addition, these results indicate that mixture effects are not simply a general consequence of PD as estimated by neutral molecular genetic variation. Instead, mixture effects depended upon the particular phylogenetic group, the amount of diversity present in a given mixture, and the response variable (i.e., plant biomass or survival). Although it is accepted that evolutionary history plays an important role in generating patterns of biodiversity, our results also identify evolutionary history as a determinant of plant biomass by showing that different phylogenetic groups can show different responses to increasing species richness and phylogenetic diversity.

Supplemental Information

Supplemental Information 1 Manuscript data filed.

Data supporting the paper’s conclusions, made available for public use.

Click here for additional data file.

Appendix S1 Effects of random species sampling on expected values in mixtures.

The following table shows the percent difference between the expected value for (1) our actual, randomly constructed mixtures and (2) hypothetical mixtures in which all species were represented equally at each level of diversity. Positive values show that the random selection picked species with above-average monoculture values for a given trait, while negative values show the opposite. The largest deviation was 2.31% and all but one of the values are less than 1.00%.

Click here for additional data file.

Thanks to University of Tasmania for use of facilities and Naeko Omomo, Camilla Bloomfield, and Justin Bloomfield for help in the greenhouse.

Additional Information and Declarations

Competing Interests

Author Contributions

Joseph Bailey is an Academic Editor for PeerJ.

Mark A. Genung analyzed the data, wrote the paper, prepared figures and/or tables, reviewed drafts of the paper.

Jennifer A. Schweitzer and Joseph K. Bailey conceived and designed the experiments, performed the experiments, analyzed the data, contributed reagents/materials/analysis tools, wrote the paper, prepared figures and/or tables, reviewed drafts of the paper.

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
