# Peer review of "Evolutionary history determines how plant productivity responds to phylogenetic diversity and species richness"

_PeerJ, doi:10.7717/peerj.288_

## Round 0.1 · original submission · Major Revisions

dear dr Genung,

Your manuscript has now been reviewed by four (!) reviewers. As you can see from their reports, they are all positive about your work, but three of the four reviewers have also made serious comments about the experimental design and the analyses. I fully understand that you cannot redo the experiment, it can be better described though, and the suggestions for different types of analyses can be incorporated in a revised version.

I fully agree with two of the reviewers that you should incorportate into your analyses that multiple plants came from the same pot, eg by inlcuding pot as a random factor in a mixed model.
Please note that one reviewer added comments into your manuscript. You should receive these comments in a separate pdf file,

I invite you to adjust the manuscript by incorporating the comments of the four reviewers and the manuscript can become acceptable for publicaton following a major revision. (the reason that there are four reviewers for your manuscript is that I contacted more potential reviewers at the same time, and then several of them accepted the invitation more or less at the same time)
Please also explain how you have dealt with the comments in a rebuttal.
I understand that it will be a bit more work than usual to adjust the manuscript based on four different sets of comments but I believe that the comments made by the reviewers are very useful and will improve the manuscript.
Good luck with the revision,

kind regards,

Martijn

Reviewer 1 ·

Basic reporting

It is well written.

Experimental design

Seems robust

Validity of the findings

Seems robust

Additional comments

The paper by Genung et al. examines how phylogenetic diversity and richness affect several performance and function measures within the Eucalyptus genus. The impetus for the study is a strong one, and the implications are interesting. I believe that this study will interest ecologists broadly.

I really only have a couple of minor concerns. First is the notion of PD as categorical bins. The ideal of within subgenera being equal evolutionarily depends on how long species within each subgenus have been independently evolving. Thus one sub-genus pairing could be 800k years separate and another 1.6 million year. We may have different expectations. The other comparison issue is how divergent species are in trait space, but the authors deal with this in the discussion.

Another concern was the complexity of the methodology, which doesn't seem to influences my reading of the results. I reread lines 145-169 several times, I still don't quite get what the concerns are and what was being done differently. Lines like "..should not be used to quantitatively compare how SR and PD affect plant performance", or "We found that the random sampling of species had minimal effects …on expected values in the mixture plots", leave me scratching my head.

Other than that, the manuscript is pretty good. I think that both the introduction and discussion set up/discuss the problem and implications very well.

Reviewer 2 ·

Basic reporting

The authors need to report more information on the experimental design and on the analysis. In particular it would be good to have a detailed description of the experimental design, perhaps in a supplementary table. What exactly were the compositions of the mixtures? How many replicates of the different combinations of species richness x subgenus richness? Were the 86 mixture pots all different compositions or was there any replication of composition? Does the n=34 (line 114) imply only 34 different compositions?

When did the two subgenera of eucalypts diverge evolutionarily? This information would help in assessing whether the phylogenetic diversity of the two subgenus mixes is really substantially higher than a one subgenus mix, it is stated that there is a "large" genetic variation between 1 and 2 subgenus mixes (line 226) but it would be good to know how large this variation is.
It would, of course, be much better if the authors had continuous measures of phylogenetic diversity rather than the rather crude approach of counting 1 or 2 subgenera; however I imagine it's not possible to get these data?

It should also be clearly specified that the analysis was done at the plant level (I'm assuming it was because it is stated in table 1 that n=497) not at the pot level. Also see below because I think this analysis is not valid.

Experimental design

I imagine that the experiment was not really designed to test the separate effects of phylogenetic and species diversity, which makes the analysis challenging and I don't think it's correct currently.

The most serious problem is that the analysis appears to be conducted at the plant level and the tree seedlings seem to be treated as if they are independent replicates of the different treatments, in table 1, n=497 for the two models and this seems to be the number of plants in the experiment (line 115). If this is correct, and the analysis contains one value per plant, then it is not valid as it needs to include a random effect for pot, to account for the non-independence between seedlings growing in the same pot. There is another source of non-independence which is that the compositions are replicated. At least the monocultures are (line 113) it is not possible to tell if the mixture compositions are replicated, see above. It is therefore necessary to also fit a random effect for composition. Finally there should be a random effect for "species" to account for mean differences in height and stem diameter between the 26 species.

I also think the modelling of the fixed effects is not correct and fitting separate models for phylogenetic diversity and species richness is really not an elegant solution. The most serious issue is that the phylogenetic diversity term contains the contrast between monocultures and mixtures, which makes it impossible to tell if it is really phylogenetic diversity rather than differences in species number driving the effects. I therefore don't think it represents the "effect of phylogenetic diversity averaging across species richness" (line 148).

I would suggest doing a mixed model analysis with fixed effects for 1) species richness 2) the number of subgenera (1 or 2) 3) subgenus identity and 4) any interactions that can be fitted between subgenus identity and other factors; it's a bit hard to know exactly what is possible without more information on the design.
If the authors really want to do two separate analyses, so that they can fit interactions between subgenus identity and the other factors, then they should at least fit the contrast between monocultures and mixtures as a term in the PD analysis. Fitting this term alongside the contrast between 1 and 2 subgenus mixes would mean that the "PD" term would not include the differences between monocultures and mixtures and would represent only the contrast between 1 and 2 subgenus mixes.
Random effects for all models would be 1) pot, 2) composition and 3) species.

Additionally I would suggest doing an analysis at the pot level to test whether diverse communities perform better than less diverse ones, in terms of average height and diameter. The current analysis tests whether tree seedlings grow better in diverse communities. This analysis would have terms for 1) species richness 2) 1 or 2 subgenus mixtures 3) subgenus identity. If one average value was used per composition, this would be a fixed effect only model. It would not be possible to test for interactions between subgenus identity and diversity here.

Validity of the findings

I don't think this can really be assessed without first redoing the analysis and providing more information on the design of the experiment.

Additional comments

In general I like the topic of the study and the paper is well written but I think there are some serious flaws in the analysis, see above. The manuscript would also be greatly improved if the authors could use a continuous measure of phylogenetic diversity rather than only the contrast between 1 and 2 subgenus mixtures. I also think it would be good to analyse mean values for the communities, see above, rather than just looking at the performance of individuals in different communities. Much of the introduction does deal with the relationship between diversity and ecosystem functioning and so it would be nice to test this. In conclusion I suggest that the authors re-analyse the data and revise the paper.

Minor comments:
"Plant-insect interactions" are mentioned on line 227 but nowhere else, perhaps cut this?
Line 81 "subgenera" should be "subgenus"
The x-axis in figure 1 should be changed, it would be clearer if it had tick marks at 1, 3 and 6 only, as these were the species richness levels involved.

·

Basic reporting

These authors address an important issue in the area of biodiversity and ecosystem function, specifically the degree to which experiments involving only manipulating species richness, without regard to phylogenetic relatedness and history, may be misinterpreting the critical components of biodiversity responsible for increasing the rates of ecosystem processes.

They attempt to do this using two subgenera of the large genus Eucalyptus in an experiment with three levels of diversity: 1, 3 and 6 species, with sub-treatments drawn from either a single subgenus, or both subgenera. While this is an important issue, and their experimental approach is appropriate, there are several weaknesses that must be addressed before the paper can resolve the questions it addresses.

1) The experiment involves tree seedlings that were germinated under controlled conditions and transplanted into pots with 6 individuals per pot after 10 weeks of growth and then grown for another 50 days (19 July through 13 Sept 2011) during which time measurements were taken. This is an extremely short time on which to draw conclusions about the effect of diversity (species richness or phylogenetic diversity) on tree species productivity. See attached abstract from Duff et al. 1983 suggesting Symphyomyrtus species initially grew faster than Monocalyptus species (synonomous with Eucalyptus in this paper?) then over time were suppressed by Monocalyptus species. They specifically contrasted the results of glasshouse experiments with results observed in the field.

The authors present insufficient information to evaluate the results of the experiments
A) What is the diameter of the 40 L pots? B) How much surface area did each individual have? C) How big were the seedlings when they were transplanted into the 40L pots? D) How big were the seedlings at the end of 50 days of growth and intra-individual interactions?

One could question the relevance of this experiment to the effects of biodiversity since all the seedlings were grown under identical monoculture conditions for 10 weeks, prior to being growth together in the diversity treatments in the greenhouse for 7 weeks, and then outside, apparently for another 7 weeks. The exact dates at which treatments were applied would be useful, as would weather data for the 7 weeks spent outside, as well as temperature and light levels in the glasshouse.

While one should never underestimate species in the genus Eucalyptus, the results of this experiment are difficult to believe. Figure 1B and 2B indicate extremely large stem diameters (5 and 10 cm) for tree seedlings with 4 months (or 6 months?) of growth. If these values represent the mean of the 6 individuals in a pot, it is astounding. If it represents the sum of the diameters of the individuals in the pots, the authors have created a novel and worthless metric that completely obscures the relative productivity of the various treatments.

Perhaps the potting mix had extremely high nutrient levels, which are probably much higher than the species would encounter in the field. 3 kg / m3 seems like a lot of fertilizer.

Regardless of what these height and diameter measurements are, the relevant metric is the biomass of each of the 6 individuals in each pot. If these data were not collected, then this paper is not publishable.


2) The paper does not distinguish between functional and phylogenetic diversity (other than citing a paper by Naeem et al. from 2011, which apparently failed to characterize these two attributes of diversity effectively.

The bottom line is that this paper cannot draw any legitimate conclusions about the effects of phylogenetic diversity without also analyzing the patterns of functional diversity within and between the two subgenera. It may be that the effects of phylogenetic diversity are caused by the patterns of functional differences within and between the two subgenera. This possibly must be clearly eliminated before “evolutionary history” per se can be invoked as a possible explanation. I doubt that there is some mysterious “evolutionary effect” that is independent of physiological and functional differences between the species.

The methods (lines 151 – 172) used to “account for the composition of species at each level of species richness... and phylogenetic diversity” (line 152) are not clearly motivated nor clearly described. What is the “proportional representation of each species in each mixture” (line 158)? Is this the planted proportion (presumably 2/6 in the 3-species treatments and 1/6 in the 6-species treatments) OR the proportion of each species at the end of the growing period? If the proportion of each species was measured as biomass and compared to the expected biomass based on the monocultures (1/3 or 1/6 for the 3 and 6 spp treatments), this would allow analysis of the interaction among the species, whether “overyielding” occurred, and which species contributed to the overyielding.

It is also essential to show information on the sizes of all the species at the end or 10 weeks growth in glasshouse flats, and prior to being transplanted into the mixtures (and monocultures).

The results that Symphyomyrtus showed a strong height response in mixtures as compared to monocultures suggests a positive response to release from intraspecific competition. However, the greater maximum height and diameter of Eucalyptus monocultures compared to Symphyomyrtus monocultures does not seem consistent with this.

Is there overyielding in any of the treatments? Which species or species combinations are responsible for the overyielding? Depending on the leaf size and orientation of the two subgenera, there might be simple niche partitioning, or perhaps not.

If there are any solid results from this experiment, they must be demonstrated by a detailed, species by species analysis of growth responses in relation to quantitative traits related to growth.

Such traits would include seed size (mass), seedling growth rates plus height and diameter at specific ages, root:shoot ratios, maximum size attained in the experiment, as well as leaf properties such as those typically measured in studies of the “Leaf Economic Spectrum” including size, mass per unit area, nitrogen content, C:N ratio, and ideally photosynthetic and respiration rates.

What are the physiological, life history, and functional differences between these two subgenera? Clearly they have very different successional dynamics in natural forests (Duff et al. 1983). What is the range of variation in specific traits in each of the two subgenera? How would this influence the results of random-selection BEF experiments such as those reported here?

The seedling mortality rates in this experiment are consistent with the observations of Duff et al.

The properties of Symphyomyrtus mentioned in lines 217-219 suggest there are major physiological and functional differences between the subgenera that must be addressed to understand the results of these experiments.

Experimental design

No Comments

Validity of the findings

No Comments

Additional comments

Given the short duration and equivocal results of this experiment, if it is to have an impact in this field it must provide a detailed analysis of which species contributed to which results and why they did so. There must be a table listing all of the species used in the experiment, their known properties, and their responses to the various numbers and types of species with which they were grown.

It must be possible to demonstrate WHY phylogeny is important for determining plant biomass. If it is important, it will be manifested in the types, range, and variability of functional traits associated with different phylogenetic groups. Hopefully, the authors can do this.



Michael Huston
Department of Biology
Texas State University
San Marcos, Texas




line 81 “from only one subgenus” not “subgenera”

line 129 – what is the representation of each subgenus in the 6 species mixtures? 3 of each? or 5 of one and 1 of the other? Hopefully 3 and 3. This should be specified.

The legends of Figs. 1 and 2 both refer to biomass, which is the preferred measure of plant growth and productivity, but no data on biomass are presented.


Important Reference::
The occurrence of mixed stands of the Eucalyptus subgenera Monocalyptus and Symphyomyrtus in south-eastern Tasmania
G. A. DUFF, J. B. REID, W. D. JACKSON DOI: 10.1111/j.1442-9993.1983.tb01337.x

Australian Journal of Ecology
Volume 8, Issue 4, pages 405–414, December 1983


Abstract

Mixed stands of eucalypts were examined on several sites in south-eastern Tasmania. Species from the subgenus Monocalyptus (E. puchella and E. obliqua) became dominant over species from the subgenus Symphyomyrtus (E. viminalis and E. globulus) as the age of the stand increased. In glasshouse trials, Symphyomyrtus species were initially at an advantage since they were more opportunistic than Monocalyptus species owing to their more rapid and even germination and higher initial growth rates. In addition, the Monocalyptus species E. pulchella showed a greater tendency to regenerate vegetatively than the Symphyomyrtus species E. viminalis and E. ovata.

Reviewer 4 ·

Basic reporting

Fine.

Experimental design

Fine.

Validity of the findings

1. There is a problem in the data analysis. The plants were grown in pots and multiple plants were measured per pot. Based on my interpretation of the M&M and results sections the hierarchical nature of the dataset was not accounted for in the models. If that is the case, then the analysis is pseudoreplicated and the conclusions cannot be trusted. This needs to be remedied before acceptance, in my opinion.

2. The description of the exact number of replicates in each treatment (Species richness levels, phylogenetic diversity levels) is missing and cannot be deduced as degrees of freedom are not reported (denominator d.f. are not used in likelihood-ratio tests).

3.According to PeerJ’s guidelines “The data […] must be provided or made available in an acceptable […] repository”, I have not seen this in this ms.

Additional comments

1. You argue that these experimental manipulations are ecologically relevant. Personally I have some doubts that it happens frequently that 6 species of Eucalyptus grow in a 30L volume of soil. The cited references were not accessible to me, and I suspect this could be the case for more colleagues. In that light, I would appreciate it if the arguments for the ecological relevance are developed a bit more. For a start, I want to see a list with the names of the species used.

2. In addition, what I am missing is a mechanistic hypothesis that explains why you, a priori, expected the two subgenera to produce different diversity-functioning relations. In the introduction there is only a very general account. This is not a big problem, and your data are interesting without it, but I have a hard time imagining how plants ‘sense’ the surrounding (phylogenetic/species) diversity, except through their traits; e.g. in the ways that thickness and positioning of leaves alters the light environment etc. (see final sentence of your discussion).

3. I’ve put further comments in the PDF. I was assured by Peter Binfield that he will forward these comments to you directly (there is no way to upload these to the website).

---

## Round 0.2 · Minor Revisions

dear Dr Genung,

I have not sent back your manuscript to the reviewers, but instead, carefully read your response to the comments of the four reviewers myself, and have read the revised version of your manuscript, submitted to PeerJ. In my opinion, you have clearly and appropriately replied to the comments of the reviewers. I agree with the reviewers that an approach using phylogenetic distance rather than using PD as categorial would have been more insightfull, but I am convinced by your response that using such phylogenetic information is not possible in your study.

Regarding the other main issue: the statistical analysis, and whether you can use data from multiple plants within a single pot, that is resolved in the revised version using pot identity as random factor as suggested. I have a small concern though. Based on the programmes I am familiar with that can be used for REML analyses, I thought these analyses produce Wald-type F values or t-values. You provide F values, please confirm this is the correct output from REML. Further, as pot is used as random factor, the error degrees of freedom should be related to the pot number and not plant number. Please provide error df in your table.

The other issues are resolved appropriately. Please could you reply to my minor comment regarding the statistics and correct the table appropriately. If that is done satisfactory the ms can then be published.

best regards,

Martijn Bezemer

---

## Round 0.3 · accepted · Accept

dear authors,

I am happy with the changes made to the manuscript, and propose to accept this manuscript for publication in PeerJ.

Congratulations!

best regards,

Martijn Bezemer